# Disentangling Content and Style via Unsupervised Geometry Distillation

**Wayne Wu**
SenseTime Research & THU & NTU
wuwenyan@sensetime.com

**Kaidi Cao**
Stanford University
kaidicao@cs.stanford.edu

**Cheng Li**
SenseTime Research
chengli@sensetime.com

**Chen Qian**
SenseTime Research
qianchen@sensetime.com

**Chen Change Loy**
Nanyang Technological University
ccloy@ntu.edu.sg

## Abstract

It is challenging to disentangle an object into two orthogonal spaces of content and style since each can influence the visual observation differently and unpredictably. It is rare for one to have access to a large number of data to help separate the influences. In this paper, we present a novel framework to learn this disentangled representation in a completely *unsupervised* manner. We address this problem in a two-branch Autoencoder framework. For the structural content branch, we project the latent factor into a soft structured point tensor and constrain it with losses derived from prior knowledge. This constraint encourages the branch to distill geometry information. Another branch learns the complementary style information. The two branches form an effective framework that can disentangle object's content-style representation without any human annotation. We evaluate our approach on four image datasets, on which we demonstrate the superior disentanglement and visual analogy quality both in synthesized and real-world data. We are able to generate photo-realistic images with $256 \times 256$ resolution that are clearly disentangled in content and style.

## 1 Introduction

Content and style are the two most inherent attributes that characterize an object visually. Computer vision researchers have devoted decades of efforts to understand object shape and extract features that are invariant to geometry change (Huang et al., 2007; Thewlis et al., 2017; Zhang et al., 2018; Rocco et al., 2018). Learning such disentangled deep representation for visual objects is an important topic in deep learning.

The main objective of our work is to disentangle object's style and content in an *unsupervised* manner. Achieving this goal is non-trivial due to three reasons: 1) Without supervision, we can hardly guarantee the separation of different representations in the latent space. 2) Although some methods like InfoGAN (Chen et al., 2016) are capable of learning several groups of independent attributes from objects, attributes from these unsupervised frameworks are uninterpretable since we cannot pinpoint which portion of the disentangled representation is related to the content and which to the style. 3) Learning structural content from a set of natural real-world images is difficult.

To overcome the aforementioned challenges, we propose a novel two-branch Autoencoder framework, of which the structural content branch aims to discover semantically meaningful structural points (*i.e.*, $y$ in Fig 2) to represent the object geometry, while the other style branch learns the complementary style representation. The settings of these two branches are asymmetric. For the structural content branch, we add a layer-wise softmax operator to the last layer. We could regard

Figure 1: **Walking in the disentangled representation space:** Three cat faces in the bounding box are from real data while others are interpolated through our learned representations.

this as a projection of a latent content to a soft structured point tensor space. Specifically designed prior losses are used to constrain the structured point tensors so that the discovered points have high repeatability across images yet distributed uniformly to cover different parts of the object. To encourage the framework to learn a disentangled yet complementary representation of both content and style, we further introduce a Kullback-Leibler (KL) divergence loss and skip-connections design to the framework. In Fig. 1, we show the latent space walking results on cat face dataset, which demonstrates a reasonable coverage of the manifold and an effective disentanglement of the content and style space of our approach.

Extensive experiments show the effectiveness of the proposed method in disentangling the content and style of natural images. We also conduct qualitative and quantitative experiments on MNIST-Color, 3D synthesized data and several real-world datasets which demonstrate the superior performance of our method to state-of-the-art algorithms.

## 2 METHODOLOGY

The architecture of our model is shown in Fig. 2. In the absence of annotation on the structure of object content, we rely on prior knowledge on how object landmarks should distribute to constrain the learning and disentanglement of structural content information. Our experiments show that this is possible given appropriate prior losses and learning architecture. We first formulate our loss function with special consideration on prior. Specifically, we follow the VAE framework and assume 1) the two latent variables $z$ and $y$, which represent the style and content, are generated from some prior distributions. 2) $x$ follows the conditional distribution $p(x|y, z)$. We start with a Bayesian formulation and maximize the log-likelihood over all observed samples $x \in X$.

$$
\begin{aligned}
\log p(x) &= \log p(y) + \log p(x|y) - \log p(y|x) \\
&\geq \log p(y) + \log \int p(x, z|y) \, \mathrm{d}z \\
&\geq \log p(y) + \mathbb{E}_q \log \frac{p(x, z|y)}{q(z|x, y)} \\
&= \log p(y) + \mathbb{E}_q \log \frac{p(x|y, z)p(z|y)}{q(z|x, y)}.
\end{aligned}
\tag{1}
$$

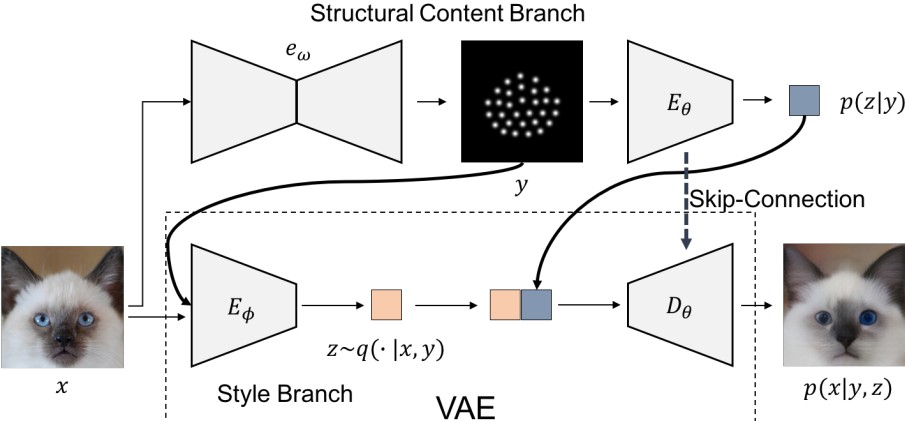

Figure 2: **Architecture:** Our framework follows an Autoencoder framework. It contains two branches: 1) the structural content branch forces the representation into a Gaussian spatial probability distribution with an hourglass network $e_\omega$. 2) the style branch $E_\phi$ learns a complementary style representation to the content.

Equation 1 learns a deterministic mapping $e(\cdot; \theta)$ from $x$ to $y$, which we assume $y$ is following a Gaussian distribution over $\mathcal{N}(e(x; \omega), \Sigma)$. Term $-\log p(y|x)$ is non-negative. In the second line of the equation, we start to consider the factor $z$. Similar to VAE, we address the issue of intractable integral by introducing an approximate posterior $q(y, z|x; \phi)$ to estimate the integral using evidence lower bound (ELBO). By splitting the $p(x|y, z)$ from the second term of the last expression, we obtain our final loss as,

$$\mathcal{L}(x, \theta, \phi, \omega) = -\log p_\omega(y) - \mathbb{E}_{q_{\phi(z|x,y)}} \log p_\theta(x|y, z) + \text{KL}(q_{\phi(z|x,y)}(z|x, y) || p_\theta(z|y)). \quad (2)$$

The first term is about the prior on $y$. The second term describes the conditional distribution of $x$ given all representation. Ideally, if the decoder can perfectly reconstruct the $x$, the second term would be a delta function over $x$. The third term represents the Kullback-Leibler divergence between approximate. In the rest of this paper we name these three terms respectively as prior loss $L_{\text{prior}}$, reconstruction loss $L_{\text{recon}}$ and KL loss $L_{KL}$.

## 2.1 PRIOR LOSS

We firstly formulate the content representation $y$ as a soft latent structured point tensor. Then, a re-projecting operator is applied here to force $y$ to lie on a Gaussian spatial probability distribution space. Following the notations from Newell et al. (2016), we denote the direct outputs of the hourglass network $e_\omega$ as landmark heatmaps $h$, and each channel of which represents the spatial location of a structural point. Instead of using max activations across each heatmap as landmark coordinates, we weighted average all activations across each heatmap. We then re-project landmark coordinates to spatial features with the same size as heatmaps by a fixed Gaussian-like function centered at predicted coordinates with a fixed standard deviation. As a result, we obtain a new tensor $y$ with the structure prior on content representation.

Nevertheless, we find that training the structural content branch with general random initialization tend to locate all structural points around the mean location at the center of the image. This could lead to a local minimum from which optimizer might not escape. As such, we introduce a *Separation Loss* to encourage each heatmap to sufficiently cover the object of interest. This is achieved by the first term in Eq. 3, where we encourage each pair of $i^{th}$ and $j^{th}$ heatmaps to share different activations. $\sigma$ can be regarded as a normalization factor here. Another prior constraint is that we wish the structural point to behave like landmarks to encode geometry structure information. To achieve this goal, we add a *Concentration Loss* to encourage the variance of activations $h$ to be small so that it could concentrate at a single location, which corresponds to the second term in Eq. 3. It is noteworthy that some recent works have considered the prior of latent factor. Dupont (2018) proposed a Joint-$\beta$-VAE by adding different prior distribution over several latent factors to disentangle continuous and discrete factors from data. Our work differs in that we investigate a different prior to disentangle visual content and style.

$$L_{prior} = \sum_{i \neq j} \exp(-\frac{||h_i - h_j||^2}{2\sigma^2}) + \text{Var}(h) \tag{3}$$

## 2.2 Reconstruction Loss

For the second term we optimize the reconstruction loss of whole model, which will be denoted as generator $G$ in the following context. We assume that the decoder $D_\theta$ is able to reconstruct original input $x$ from latent representation $y$ and $z$, which is $\hat{x} = G(y, z)$. Consequently, we can design the reconstruction loss as $L_{\text{recon}} = \|x - \hat{x}\|_1$.

However, minimizing $L_1$ / $L_2$ loss at pixel-level only does not model the perceptual quality well and makes the prediction look blurry and implausible. This phenomenon has been well-observed in the literature of super-resolution (Bruna et al., 2016; Sajjadi et al., 2017). We consequently define the reconstruction loss as $L_{\text{recon}} = \|x - \hat{x}\|_1 + \sum_l \lambda_l \|\psi_l(x) - \psi_l(\hat{x})\|_1$, where $\psi_l$ is the feature obtained from $l$-th layer of a VGG-19 model (Simonyan & Zisserman, 2014) pre-trained on ImageNet. It is also possible to add adversarial loss to further improve the perceptual reconstruction quality. Since the goal of this work is disentanglement rather than reconstruction, we only adopt the $L_{\text{recon}}$ described above.

## 2.3 KL Loss

We model $q(z|x, y)$ as a parametric Gaussian distribution which can be estimated by the encoder network $E_\phi$. Therefore, the style code $z$ can be sampled from $q(z|x, y)$. Meanwhile, the prior $p(z|y)$ can be estimated by the encoder network $E_\theta$. By using the reparametrization trick (Kingma & Welling, 2014), these networks can be trained end-to-end. We only estimate mean value here for the stability of learning. By modeling the two distributions as Gaussian with identity covariances, the KL Loss is simply equal to the Euclidean distance between their means. Thus, $z$ is regularized by minimizing the KL divergence between $q(z|x, y)$ and $p(z|y)$.

Notice that with only prior and reconstruction loss. The framework only makes sure $z$ is from $x$ and the Decoder $D_\theta$ will recover as much information of $x$ as possible. There is no guarantee that $z$ will learn a complementary of $y$. Towards this end, as shown in Fig. 2, we design the network as fusing the encoded content representation by $E_\theta$ with the inferred style code $z$. Then, the fused representation is decoded together by $D_\theta$. Meanwhile, skip-connections between $E_\theta$ and $D_\theta$ are used to pass multi-level content information to the decoder. Therefore, enough content information can be obtained from prior and any information about content encoded in $z$ incurs a penalty of the likelihood $p(x|y, z)$ with no new information (*i.e.* style information) is captured. This design of the network and the KL Loss result in a constraint to guide $z$ to encode more information about the style which is complementary to content.

## 2.4 Implementation Detail

Each of the input images $x$ is cropped and resized to $256 \times 256$ resolution. A one-stack hourglass network (Newell et al., 2016) is used as a geometry extractor $e_\omega$ to project input image to the heatmap $y \in \mathbb{R}^{256 \times 256 \times 30}$, in which each channel represents one point-centered 2D-Gaussian map (with $\sigma = 4$). $y$ is drawn in a single-channel map for visualization in Fig. 2. Same network (with stride-2 convolution for downsampling) is used for both $E_\theta$ and $E_\phi$ to obtain style representation $z$ and the embedded content representation as two 128-dimension vectors. A symmetrical deconvolution network with skip connection is used as the decoder $D_\theta$ to get the reconstructed result $\hat{x}$. All of the networks are jointly trained from scratch end-to-end. We detail the architectures and hyperparameters used for our experiments in appendix A.

## 3 Related Work

**Unsupervised Feature Disentangle:** Several pioneer works focus on unsupervised disentangled representation learning. Following the propose of GANs (Goodfellow et al., 2014), Chen et al. (2016) purpose InfoGAN to learn a mapping from a group of latent variables to the data in an

unsupervised manner. Many similar methods were purposed to achieve a more stable result (Higgins et al., 2017; Kumar et al., 2018). However, these works suffer to interpret, and the meaning of each learned factor is uncontrollable. There are some following works focusing on dividing latent factors into different sets to enforce better disentangling. Mathieu et al. (2016) assign one code to the specified factors of variation associated with the labels, and left the remaining as unspecified variability. Similar to Mathieu et al. (2016), Hu et al. (2018) then propose to obtain disentanglement of feature chunks by leveraging Autoencoders, with the supervision of some same/different class pairs. Shu et al. (2017) rely on a 3DMM model together with GANs to disentangle representation of face properties. Dupont (2018) divides latent variable into discrete and continuous one, and distribute them in different prior distribution. In our work, we give one branch of representation are more complicated prior, to force it to represent only the shape information for the object.

**Supervised Pose Synthesis:** Recently the booming of GANs research improves the capacity of pose-guided image generation. Ma et al. (2017) firstly try to synthesize pose images with U-Net-like networks. Several works soon follow this appealing topic and obtain better results on human pose or face generation. Esser et al. (2018) applied a conditional U-Net for shape-guided image generation. (Kossaifi et al., 2018; Qiao et al., 2018) incorporat geometric information into the image generation process. Nevertheless, existing works rely on massive annotated datas, they need a strong pre-trained landmark estimator, or treat landmarks of a object as input.

**Unsupervised Structure Learning:** Unsupervised learning structure from objects is one of the essential topics in computer vision. The rudimentary works focus on keypoints detection and learning a strong descriptor to match (Thewlis et al., 2017; Rocco et al., 2018). Recently, Jakab et al. (2018) and Zhang et al. (2018), show the possibility of end-to-end learning of structure in Autoencoder formulations. Shu et al. (2018) follow the deformable template paradigm to represent shape as a deformation between a canonical coordinate system and an observed image. Our work is diffenent from the aforementioned methods mainly in the explicitly learned complementary style representations and the formulation in a two-branch VAE framework which leads to a clear disentanglement.

# 4 EXPERIMENTS

## 4.1 EXPERIMENTAL PROTOCOL

**Datasets**: We evaluate our method on four datasets that cover both synthesized and real world data: 1). MNIST-Color: we extend MNIST by either colorizing the digit (MNIST-CD) or the background (MNIST-CB) with a randomly chosen color following Gonzalez-Garcia et al. (2018). We use the standard split of training (50k) and testing (10k) set. 2). 3D Chair: Aubry et al. (2014) offers rendered images of 1393 CAD chair models. We take 1343 chairs for training and the left 50 chairs for testing. For each chair, 12 rendered images with different views are selected randomly. 3). Cat & Dog Face, we collect 6k (5k for training and 1k for testing) images of cat and dog from YFCC100M (Kalkowski et al., 2015) and Stanford Dog (Khosla et al., 2011) datasets respectively. All images are center cropped around the face and scaled to the same size. 4). CelebA: it supplies plenty of celebrity faces with different attributes. The training and testing sizes are 160K and 20K respectively.

**Evaluation Metric**: We perform both qualitative and quantitative evaluations to study the disentanglement ability and generation quality of the proposed framework: 1). **Qualitative:** we provide four kinds of qualitative results to show as many usages of the disentangled space as possible, *i.e.* conditional sampling, interpolation, retrieval, and visual analogy. 2). **Quantitative:** we apply several metrics that are widely employed in image generation (a) Content consistency: content similarity metric (Li et al., 2017) and mean-error of landmarks (Bulat & Tzimiropoulos, 2017). (b) Style consistency: style similarity metric (Johnson et al., 2016) (c). Disentangled ability: retrieval recall@K (Sangkloy et al., 2016). (d). Reconstruction and generation quality: SSIM (Wang et al., 2004) and Inception Score (Salimans et al., 2016).

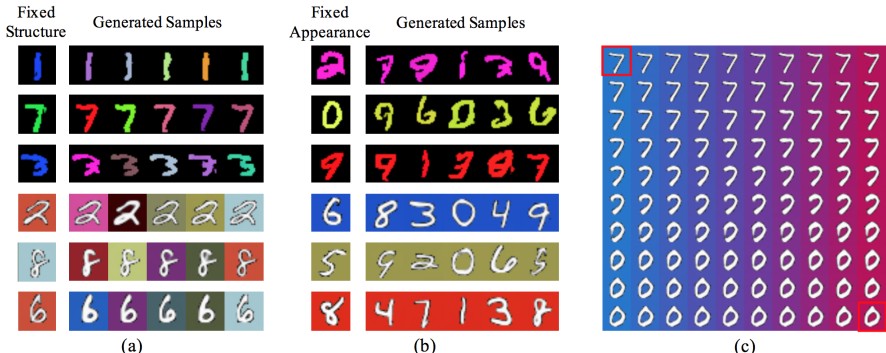

Figure 3: **Conditional generation results:** (a) Walking in the style space with fixed content. (b) Walking in the content space with fixed style. (c) A visualization of the disentangled space by linear interpolation. The content is smoothly changed in row-wise and the style is changed by each column.

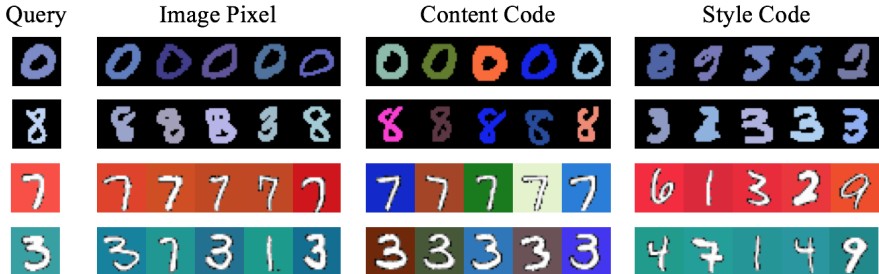

Figure 4: Random chosen 4 query images and the corresponding 5 nearest-neighbors are illustrated, which are retrieved with image pixel, content code, style code respectively.

## 4.2 RESULTS ON SYNTHESIZED DATASETS

**Diverse Generation.** We first demonstrate the diversity of conditional generation results on MNIST-Color with the successfully disentangled content and style in Fig. 3. It can be observed that, given an image as a content condition, same digit information with different style can be generated by sampling the style condition images randomly. While given an image as style condition, different digits with the same color can be generated by sampling different structural conditional images. Note that the model has no prior knowledge of the digit in the image as no label is provided, it effectively learns the disentanglement spontaneously.

**Interpolation.** In Fig. 3, the linear interpolation results show reasonable coverage of the manifold. From left to right, the color is changed smoothly from blue to red with interpolated style latent space while maintaining the digit information. Analogously, the color stays stable while one digit transforms into the other smoothly from top to down.

**Retrieval.** To demonstrate the disentangled ability of the representation learned by the model, we perform nearest neighbor retrieval experiments following Mathieu et al. (2016) on MNIST-Color. With content and style representation used, both semantic and visual retrieval can be performed respectively. The Qualitative results are shown in Fig. 4. Quantitatively, We use a commonly used retrieval metric Recall@K as in (Sangkloy et al., 2016; Pang et al., 2017), where for a particular query digit, Recall@K is 1 if the corresponding digit is within the top-K retrieved results and 0 otherwise. We report the most challenging Recall@1 by averaging over all queries on the test set in Table 2 (a). It can be observed that the content representation shows the best performance and clearly outperforms image pixel and style representation. In addition to the disentangled ability, this result also shows that the content representation learned by our model is useful for visual retrieval.

**Visual Analogy.** The task of visual analogy is that the particular attribute of a given reference image can be transformed to a query one (Reed et al., 2015). We show the visual analogy results on

MNIST-Color and 3D Chair in Fig. 5. Note that even for the detail component (*e.g.* wheel and leg of 3D chair) the content can be maintained successfully, which is a rather challenging task in previous unsupervised works (Chen et al., 2016; Higgins et al., 2017).

**Comparison.** We compare perceptual quality with the three most related unsupervised representation learning methods in Fig. 6, *i.e.*, VAE (Kingma & Welling, 2014), $\beta$-VAE (Higgins et al., 2017) and InfoGAN (Chen et al., 2016). It can be observed that from left to right, all of the three methods can rotate the chairs successfully, which demonstrates the automatic learning of disentangling the factor of azimuth on 3D Chair dataset. However, it can be perceived that the geometry shape can be maintained much better in our approach than all the other methods, owing to the informative prior supplied by our structural content branch.

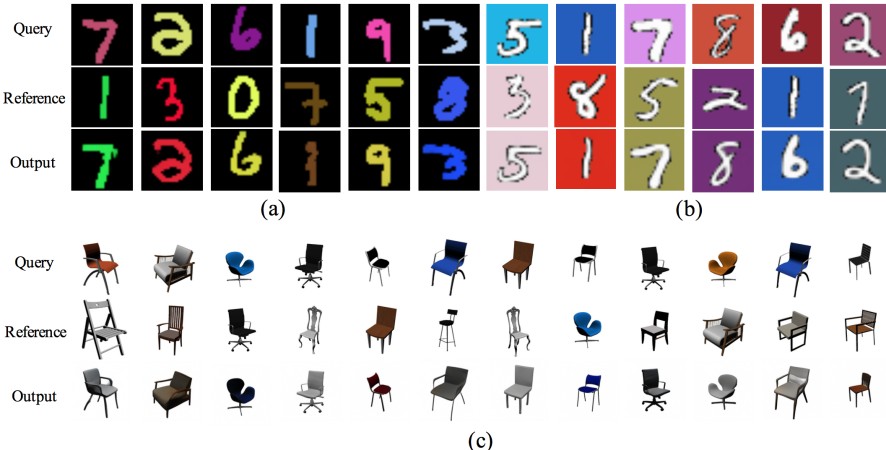

Figure 5: **Visual analogy results on synthesized datasets:** (a) MNIST-CD. (b) MNIST-CB. (c) 3D Chair. Taking the content representation of a query image and the style representation of the reference one, our model can output an image which maintains the geometric shape of query image while capturing the style of the reference image.

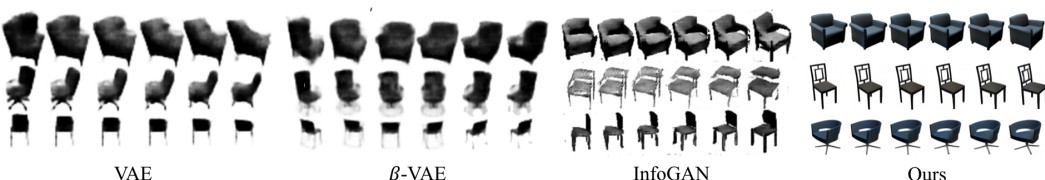

Figure 6: **Comparison to other methods.** Qualitative results of disentangling performance of VAE (Kingma & Welling, 2014), $\beta$-VAE (Higgins et al., 2017) and InfoGAN (Chen et al., 2016). We demonstrate the disentanglement ability of our method of the azimuth factor for 3D Chair dataset and much better geometry maintaining ability from left to right than state-of-the-arts.

## 4.3 RESULTS ON REAL-WORLD DATASETS

We have so far only discussed results on the synthesized benchmarks. In this section, we will demonstrate the scalable performance of our model on several real-world datasets, *i.e.*, Cat, Dog Face and CelebA. To the best knowledge of ours, there is no literature of unsupervised disentanglement before can successfully extend to photo-realistic generation with $256 \times 256$ resolution. Owing to the structural prior which accurately capture the structural information of images, our model can transform style information while faithfully maintain the geometry shapes.

Qualitative evaluation is performed by visually examining the perceptual quality of the generated images. In Fig. 8, the swapping results along with the learned structural heatmaps $y$ are illustrated on Cat dataset. In can be seen that the geometry information, *i.e.*, expression, head-pose, facial action, and style information *i.e.*, hair texture, can be swapped between each other arbitrarily. The learned structural heatmaps can be shown as a map with several 2D Gaussian points, which successfully encode the geometry cues of a image by the location of its points and supply an effective prior for

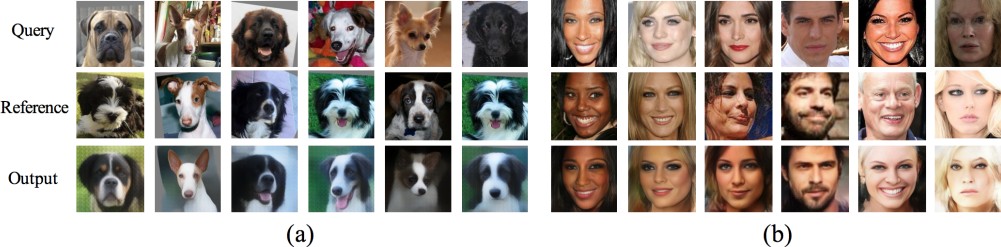

|            | (a)                          |            | (b)                          |

Figure 7: **Visual analogy results on real-world datasets:** (a) Stanford Dog. (b) CelebA. The content (*e.g.* identity, head pose and expression) of query image can be faithfully maintained while the style (*e.g.* the color of hair, beard and illumination) of reference image can be precisely transformed. As concrete examples, the output of the dog in the third column is still tongue-sticking while the hair color is changed, and in the last column of CelebA, even the fine-grain eye make-up is successfully transformed to the query image surprisingly.

| Method | Cat | | | | | CelebA | | | | |
|---|---|---|---|---|---|---|---|---|---|---|
| | Style ($\times e^{-5}$) | Content ($\times e^{-6}$) | Landmark (%) | SSIM | IS | Style ($\times e^{-5}$) | Content ($\times e^{-6}$) | Landmark (%) | SSIM | IS |
| Ours + Random | 7.700 | 1.881 | 0.051 | 0.449 | 1.968 | 5.858 | 1.693 | 0.293 | 0.532 | 1.952 |
| Zhang et al. (2018) | 6.034 | 1.812 | 0.041 | 0.389 | 1.691 | 4.571 | 1.598 | 0.272 | 0.450 | 1.627 |
| Jakab et al. (2018) | 5.963 | 1.803 | 0.038 | 0.327 | 1.529 | 4.256 | 1.602 | 0.218 | 0.411 | 1.593 |
| **Ours** | **5.208** | **1.759** | **0.030** | **0.449** | **1.968** | **3.886** | **1.529** | **0.162** | **0.532** | **1.952** |

Table 1: **Quantitative results on real-world datasets.** Disentanglement ability and generation quality comparison with state-of-the-arts on Cat and CelebA dataset. Style, content and landmark: lower is better. SSIM and IS: higher is better

the VAE network. More results of visual analogy of real-world datasets on Stanford Dog and CelebA dataset are illustrated in Fig. 7. We observe that our model can successfully generalize to various real-world images with large variations, such as mouth-opening, eye-closing, tongue-sticking and exclusive style.

For quantitative measurement, there is no standard evaluation metric of the quality of the visual analogy results for real-world datasets since ground-truth targets are absent. We propose to evaluate the content and style consistency of the analogy predictions respectively instead. We use content similarity metric for the evaluation of content consistency between a condition input $x_s$ and its guided generated images (*e.g.*, for each column of images in Fig. 8). We use style similarity metric to evaluate the style consistency between a condition input $x_a$ and its guided generated images (*e.g.*, each row of images in Fig. 8). These two metrics are used widely in image generation applications as an objective for training to maintain content and texture information (Li et al., 2017; Johnson et al., 2016).

Since content similarity metric is less sensitive to the small variation of images, we further propose to use the mean-error of landmarks detected by a landmark detection network, which is pre-trained on manually annotated data, to evaluate the content consistency. Since the public cat facial landmark annotations are too sparse to evaluate the content consistency (*e.g.* 9-points (Zhang et al., 2008)), we manually annotated 10k cat face with 18-points to train a landmark detection network for evaluation purpose. As for the evaluation of celebA, a state-of-the-art model (Bulat & Tzimiropoulos, 2017) with 68-landmarks is used.

The results on the testing set of the two real-world datasets are reported in Table 1. For each test image, 1k other images in the testing set are all used as the reference of content or style for generation, in which mean value is calculated. In the baseline "Ours + random" setting, for one test image, the mean value is calculated by sampling randomly among the generated images guided by each image. Results of two state-of-the-art unsupervised structure learning methods (Jakab et al., 2018; Zhang et al., 2018) are also reported for comparison. Content consistency is evaluated by content similarity metric and landmark detection error, while style consistency is evaluated by style similarity metric as mentioned above. Structural Similarities (SSIM) and Inception Scores (IS) are utilized to evaluate the reconstruction quality and the analogy quality. Superior performance of both content/style consistency and generation quality of our method can be obviously observed in Table 1.

### 4.4 ABLATION STUDY

To study the effects of VGG loss (Sec. 2.2) and KL loss (Sec. 2.3) of our method on generated images. We evaluate the aforementioned metrics of our method on Cat dataset. As reported in Table 2 (b), without VGG loss, the style consistency degraded slightly while the Inception Score decreased a lot for the severe blurry genration results. Without KL loss, the network has no incentive to learn the content-invariant style of representation, almost all of the metrics degraded dramatically.

| Method | Color-Digit | Color-Back |
|---------|-------------|------------|
| Pixel | 31.65 | 39.52 |
| Style | 10.25 | 15.32 |
| Content | **99.96** | **99.92** |

(a)

| Method | Style ($\times e^{-5}$) | Content ($\times e^{-6}$) | Landmark (%) | SSIM | IS |
|---------|------------|-------------|--------------|------|-----|
| Real Data | - | - | - | 1.000 | 2.004 |
| Ours w/o VGG | 5.897 | 1.762 | 0.032 | 0.435 | 1.796 |
| Ours w/o KL | 6.556 | 1.813 | 0.036 | 0.406 | 1.720 |
| **Ours** | **5.208** | **1.759** | **0.030** | **0.449** | **1.968** |

(b)

Table 2: (a) Retrieval results reported as recall@1 on MNIST-Color. (b) Ablation study on Cat dataset.

## 5 CONCLUSION

We propose a novel model based on Autoencoder framework to disentangle object's representation by content and style. Our framework is able to mine structural content from a kind of objects and learn content-invariant style representation simultaneously, without any annotation. Our work may also reveal several potential topics for future research: 1) Instead of relying on supervision, using strong prior to restrict the latent variables seems to be a potential and effective tool for disentangling. 2) In this work we only experiment on near-rigid objects like chairs and faces, learning on deformable objects is still an opening problem. 3) The content-invariant style representation may have some potentials on recognition tasks.

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

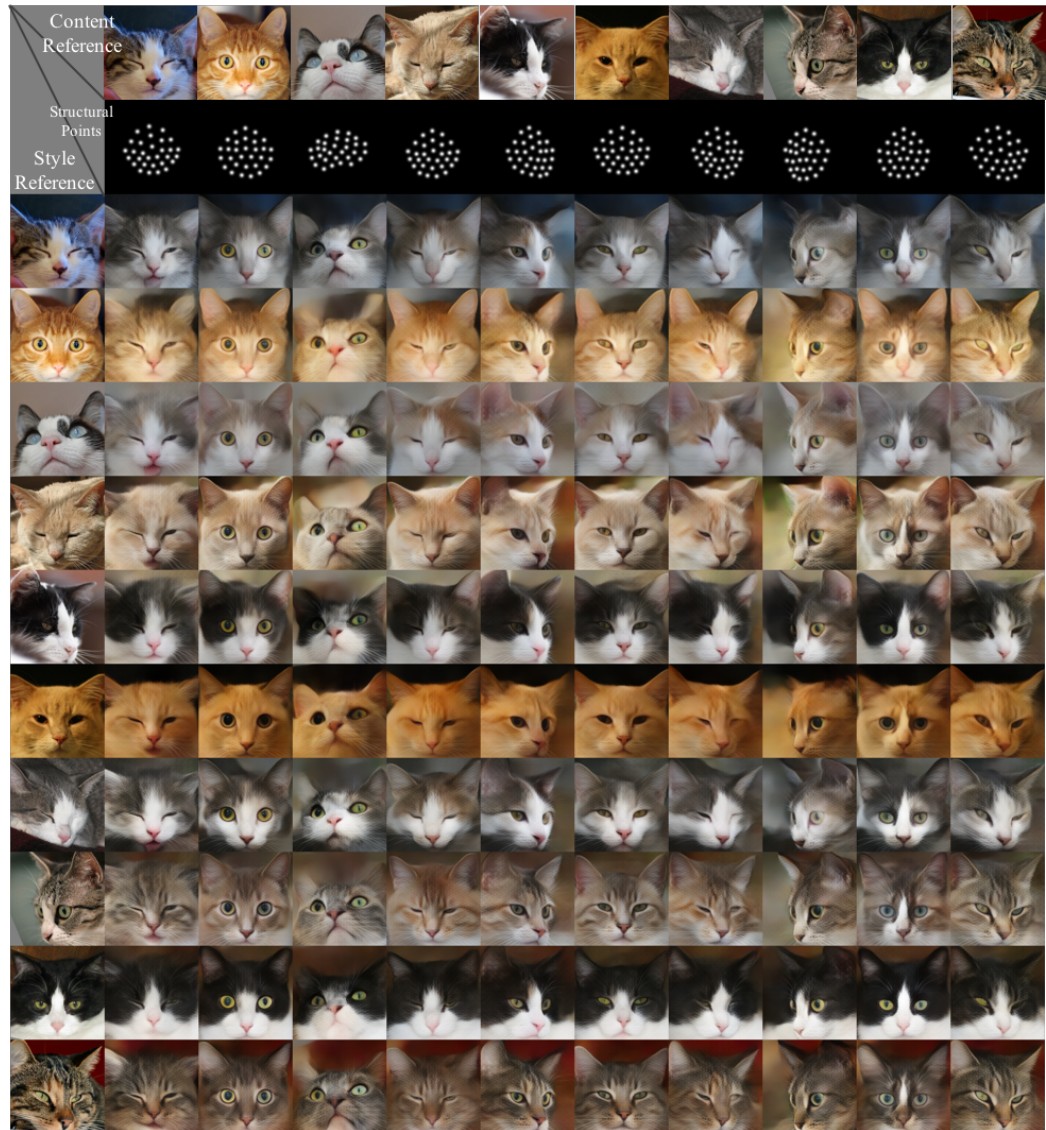

Figure 8: **A grid of content&style swapping visualization.** The top row and left-most columns are random selected from the test set. In each column, the content of the generated images are shown to be consistent with the top ones. In each row, the style of the generated images are shown to be consistent with the left-most ones.

Gary B. Huang, Vidit Jain, and Erik G. Learned-Miller. Unsupervised joint alignment of complex images. In *ICCV*, 2007.

Tomas Jakab, Ankush Gupta, Hakan Bilen, and Andrea Vedaldi. Conditional image generation for learning the structure of visual objects. *NIPS*, 2018.

Justin Johnson, Alexandre Alahi, and Li Fei-Fei. Perceptual losses for real-time style transfer and super-resolution. In *ECCV*, 2016.

Sebastian Kalkowski, Christian Schulze, Andreas Dengel, and Damian Borth. Real-time analysis and visualization of the yfcc100m dataset. In *MM Workshop*, 2015.

Aditya Khosla, Nityananda Jayadevaprakash, Bangpeng Yao, and Li Fei-Fei. Novel dataset for fine-grained image categorization. In *CVPR Workshop*, 2011.

Diederik P. Kingma and Max Welling. Auto-encoding variational bayes. In *ICLR*, 2014.

Jean Kossaifi, Linh Tran, Yannis Panagakis, and Maja Pantic. Gagan: Geometry-aware generative adversarial networks. In *CVPR*, 2018.

Abhishek Kumar, Prasanna Sattigeri, and Avinash Balakrishnan. Variational inference of disentangled latent concepts from unlabeled observations. In *ICLR*, 2018.

Yijun Li, Chen Fang, Jimei Yang, Zhaowen Wang, Xin Lu, and Ming-Hsuan Yang. Universal style transfer via feature transforms. In *NIPS*, 2017.

Liqian Ma, Xu Jia, Qianru Sun, Bernt Schiele, Tinne Tuytelaars, and Luc Van Gool. Pose guided person image generation. In *NIPS*, 2017.

Michael F Mathieu, Junbo Jake Zhao, Junbo Zhao, Aditya Ramesh, Pablo Sprechmann, and Yann LeCun. Disentangling factors of variation in deep representation using adversarial training. In *NIPS*, 2016.

Alejandro Newell, Kaiyu Yang, and Jia Deng. Stacked hourglass networks for human pose estimation. In *ECCV*, 2016.

Kaiyue Pang, Yi-Zhe Song, Tony Xiang, and Timothy M. Hospedales. Cross-domain generative learning for fine-grained sketch-based image retrieval. In *BMVC*, 2017.

Fengchun Qiao, Naiming Yao, Zirui Jiao, Zhihao Li, Hui Chen, and Hongan Wang. Geometry-contrastive gan for facial expression transfer. *arXiv preprint arXiv:1802.01822*, 2018.

Scott E. Reed, Yi Zhang, Yuting Zhang, and Honglak Lee. Deep visual analogy-making. In *NIPS*, 2015.

Ignacio Rocco, Relja Arandjelovic, and Josef Sivic. End-to-end weakly-supervised semantic alignment. In *CVPR*, 2018.

Mehdi S. M. Sajjadi, Bernhard Schölkopf, and Michael Hirsch. Enhancenet: Single image super-resolution through automated texture synthesis. In *ICCV*, 2017.

Tim Salimans, Ian J. Goodfellow, Wojciech Zaremba, Vicki Cheung, Alec Radford, and Xi Chen. Improved techniques for training gans. In *NIPS*, 2016.

Patsorn Sangkloy, Nathan Burnell, Cusuh Ham, and James Hays. The sketchy database: learning to retrieve badly drawn bunnies. In *ACM Transactions on Graphics*, 2016.

Z. Shu, E. Yumer, S. Hadap, K. Sunkavalli, E. Shechtman, and D. Samaras. Neural face editing with intrinsic image disentangling. In *CVPR*, 2017.

Zhixin Shu, Mihir Sahasrabudhe, Riza Alp Guler, Dimitris Samaras, Nikos Paragios, and Iasonas Kokkinos. Deforming autoencoders: Unsupervised disentangling of shape and appearance. In *ECCV*, 2018.

K. Simonyan and A. Zisserman. Very deep convolutional networks for large-scale image recognition. volume abs/1409.1556, 2014.

James Thewlis, Hakan Bilen, and Andrea Vedaldi. Unsupervised learning of object frames by dense equivariant image labelling. In *NIPS*, 2017.

Zhou Wang, Alan C. Bovik, Hamid R. Sheikh, and Eero P. Simoncelli. Image quality assessment: from error visibility to structural similarity. *IEEE Trans. Image Processing*, 13(4):600–612, 2004.

Weiwei Zhang, Jian Sun, and Xiaoou Tang. Cat head detection - how to effectively exploit shape and texture features. In *ECCV*, 2008.

Yuting Zhang, Yijie Guo, Yixin Jin, Yijun Luo, Zhiyuan He, and Honglak Lee. Unsupervised discovery of object landmarks as structural representations. In *CVPR*, 2018.

# A  APPENDIX

## A.1  DETAILS OF ARCHITECTURE

We use Adam with parameters $\beta_1 = 0.5$ and $\beta_1 = 0.999$ to optimise the network with a mini-batch size of 8 for 160 epochs for all datasets. The initial learning rate is set to be $0.0001$ and then decreasely linearly to 0 during training.

The network architecture used for our experiments is given in Table 3. We use the following abbreviation for ease of presentation: N=Neurons, K=Kernel size, S=Stride size. The transposed convolutional layer is denoted by DCONV.

|  | Layer | Module |
|---|---|---|
| | 1 | CONV-(N64,K4,S2) |
| | 2 | LeaklyReLU, CONV-(N128,K4,S2), InstanceNorm |
| | 3 | LeaklyReLU, CONV-(N128,K4,S2), InstanceNorm |
| | 4 | LeaklyReLU, CONV-(N128,K4,S2), InstanceNorm |
| Encoder $(E_\phi, E_\theta)$ | 5 | LeaklyReLU, CONV-(N128,K4,S2), InstanceNorm |
| | 6 | LeaklyReLU, CONV-(N128,K4,S2), InstanceNorm |
| | 7 | LeaklyReLU, CONV-(N128,K4,S2), InstanceNorm |
| | 8 | LeaklyReLU, CONV-(N128,K4,S2), InstanceNorm |
| | $\mu$ | CONV-(N128,K1,S1) |
| | 1 | CONV-(N128,K1,S1) |
| | 2 | ReLU, DCONV-(N128,K4,S2), InstanceNorm |
| | 3 | ReLU, DCONV-(N128,K4,S2), InstanceNorm |
| | 4 | ReLU, DCONV-(N128,K4,S2), InstanceNorm |
| | 5 | ReLU, DCONV-(N128,K4,S2), InstanceNorm |
| Decoder $(D_\theta)$ | 6 | ReLU, DCONV-(N128,K4,S2), InstanceNorm |
| | 7 | ReLU, DCONV-(N128,K4,S2), InstanceNorm |
| | 8 | ReLU, DCONV-(N64,K4,S2), InstanceNorm |
| | 9 | ReLU, DCONV-(N3,K4,S2), Tanh |

Table 3: Network architecture of encoder and decoder.

## A.2  MORE QUALITATIVE RESULTS

Interpolation results of 3D Chair with same arrangement as MNIST-Color is shown in Fig. 9.

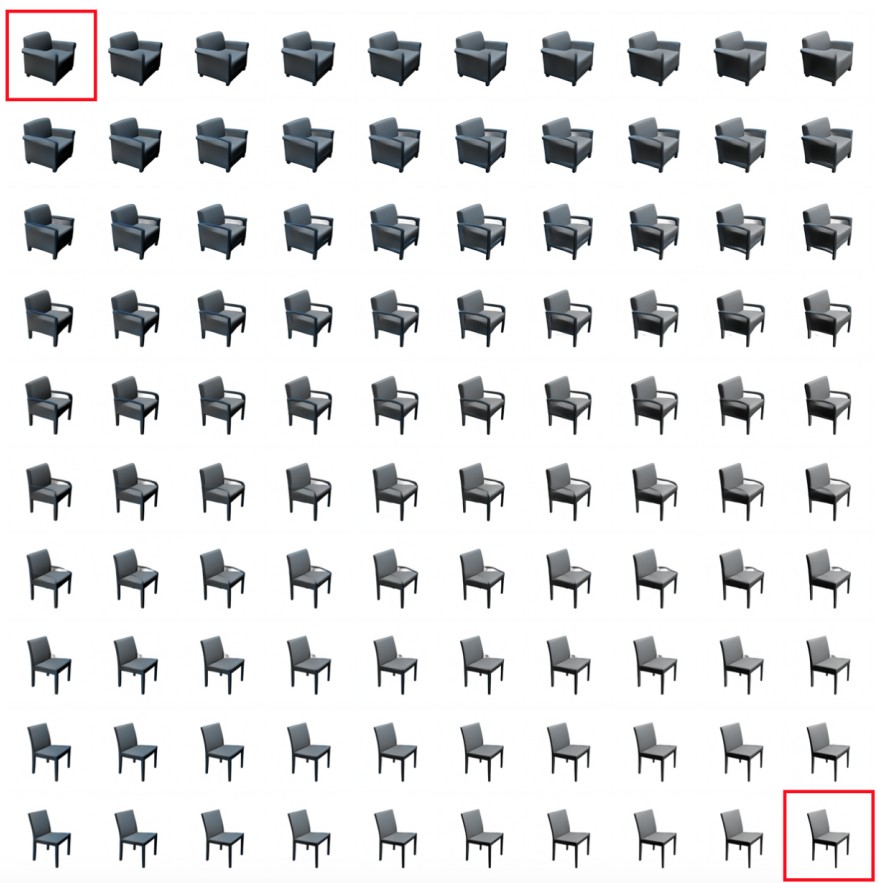

Figure 9: Interpolation results on 3D Chair.

