# OpenReview forum: "Disentangling Content and Style via Unsupervised Geometry Distillation"
_ICLR.cc/2019/Workshop/DeepGenStruct — DeepGenStruct 2019_

### Official Review · AnonReviewer1 · 2019-04-15
**Two-Branch autoencoder model for simultaneous content and style unsupervised learning**

**Rating:** 3
**Confidence:** 1

**Review:**

In this paper, a model based on autoencoder framework is proposed to disentangle object’s representation by content and style in an unsupervised manner. The model contains two branches, one for content and the other for style. The structural content branch looks at the structural points to capture the object geometry, while the style branch learns the style representation. The objective function contains the prior loss, reconstruction loss and the KL loss, which is an extension of the traditional VAE framework.

Experiments show that the proposed model can to some extent produce representations capturing both content and style. In particular, a content representation of the query image and the style representation of the reference one can output an image maintaining the geometric information of the query while having the style of the reference. In terms of quantitative results, the proposed method also outperforms existing methods wrt SSIM and IS score.

---

### Official Review · AnonReviewer2 · 2019-04-15
**interesting model, sloppy presentation with many information gaps**

**Rating:** 2
**Confidence:** 1

**Review:**

This paper proposes an unsupervised method for disentangling the content from
the style of images. I proposes a 2-branch VAE. The structure branch is
explicitly designed to capture geometric properties, and the style branch should
capture 'everything else' as style. Extensive  evaluation shows that the model
outperforms competitors qualitatively and quantitatively on real and artificial
data sets.

The paper proposes an interesting model and presents an impressive diversity
of evaluation. Unfortunately, the writing is often unclear (and the paper has a
large number of grammar errors) which makes the model description, and especially
the evaluation, very hard to follow.

Pros
------
- interesting and intuitive prior which drives content capturing
- exhaustive evaluation; good effort to capture results quantitatively
- ablation study showing benefit of different model components

Cons
------
- quality of writing; many grammar mistakes (specifically 2.3 and 2.4);
incomplete sentences (e.g., beginning of 2.3; last paragraph on page 2)
- many details left out (e.g., in model description and evaluation), model description
 is hard to follow at places
- evaluation insufficiently described; many tasks are not comprehensible form
the provided information.

Detailed Comments
------------------
- a motivation of why the task of content and style disentanglement is useful
(or interesting) would improve the paper. Relatedly, Figure 1 is very much left
out of context until the experiment section; it could be used to provide the
motivation, and should be explained in more detail in the intro, or moved to a
later spot.
- The prior loss description is very dense. The hour glass network and the
landmark mapping process should be explained in detail. It does not become clear
if this is standard methodology or a contribution of this work.
- an Explanation of hourglass networks should be added.
- the description of the KL Loss penalty formulation (second part of 2.3) is
very dense and hard to follow
- many details for the evaluation are missing, e.g., regarding SSIM and IS
scores
- The figures and tables should be rearranged so that they appear close to where
they are discussed in text. A single table should not include results from
different experiments (as done in Table 2)
- Retrieval experiments should include comparison against a baseline model and
competitive comparison models to put the proposed models' scores into context
- The Comparison evaluation (in 4.2) is insufficiently described. What is the
azimuth factor? From Figure 5 the difference in performance between the systems
is not obvious. What do the models predict? What changes along the rows?
- Last paragraph on p. 6 refers to Figure 7, you probably mean Figure 1.
- good intuitive explanation of geometry prior in last paragraph page 6 ("The
learnt structural heatmaps...") This explanation would be useful to have earlier
in the paper.
- Will the 18-point landmark annotated data be made available?
- last paragraph of page 7 is very hard to follow

---

### Decision · Program_Chairs · 2019-04-19
**Acceptance Decision**

Accept